# Current Status of Biomedical Products for Gene and Cell Therapy of Recessive Dystrophic Epidermolysis Bullosa

**DOI:** 10.3390/ijms251910270

**Published:** 2024-09-24

**Authors:** Alla Zorina, Vadim Zorin, Artur Isaev, Dmitry Kudlay, Natalia Manturova, Andrei Ustugov, Pavel Kopnin

**Affiliations:** 1Artgen Biotech, Moscow 119333, Russia; zorina@hsci.ru (A.Z.);; 2Skincell LLC, Moscow 119333, Russia; 3Department of Pharmacology, The I. M. Sechenov First Moscow State Medical University (The Sechenov University), Moscow 119991, Russia; 4Department of Pharmacognosy and Industrial Pharmacy, Lomonosov Moscow State University, Moscow 119992, Russia; 5Department of Plastic and Reconstructive surgery, Cosmetology and Cell Technologies, Pirogov Russian National Research Medical University, Moscow 117997, Russia; 6JSC Plastic Surgery and Cosmetology Institute, Moscow 125047, Russia; 7Scientific Research Institute of Carcinogenesis, N. N. Blokhin National Medical Research Center of Oncology, Moscow 115522, Russia

**Keywords:** epidermolysis bullosa, COL7A1, skin, treatment, gene and cell therapy, stem cells, gene delivery, study outcomes

## Abstract

This detailed review describes innovative strategies and current products for gene and cell therapy at different stages of research and development to treat recessive dystrophic epidermolysis bullosa (RDEB) which is associated with the functional deficiency of collagen type VII alpha 1 (C7) caused by defects in the *COL7A1* gene. The use of allogenic mesenchymal stem/stromal cells, which can be injected intradermally and intravenously, appears to be the most promising approach in the field of RDEB cell therapy. Injections of genetically modified autologous dermal fibroblasts are also worth mentioning under this framework. The most common methods of RDEB gene therapy are gene replacement using viral vectors and gene editing using programmable nucleases. Ex vivo epidermal transplants (ETs) based on autologous keratinocytes (Ks) have been developed using gene therapy methods; one such ET successively passed phase III clinical trials. Products based on the use of two-layer transplants have also been developed with both types of skin cells producing C7. Gene products have also been developed for local use. To date, significant progress has been achieved in the development of efficient biomedical products to treat RDEB, one of the most severe hereditary diseases.

## 1. Introduction

Recessive dystrophic epidermolysis bullosa (RDEB) is a type of congenital epidermolysis bullosa (EB), a disease included in the phenotypically and genetically heterogeneous group of genodermatoses [1,2]—which arise due to hereditary defects in structural proteins providing dermal–epidermal junctions (DEJs) in the basement membrane zone (BMZ) and are clinically manifested by skin fragility, the formation of blisters, chronic wounds, and damaged mucous linings.

According to the level of blister formation within the BMZ and the localization of affected C7, congenital EB is subdivided into four major types [2,3,4], namely the following: (1) simplex EB characterized by intraepidermal blistering; (2) junctional EB with blistering in the lamina lucida; (3) dystrophic EB (DEB) that in turn is subdivided into dominant dystrophic epidermolysis bullosa (DDEB) and recessive dystrophic epidermolysis bullosa (RDEB) and is located in the sub-lamina densa; and (4) Kindler syndrome, a rare type of EB characterized by mixed levels of tissue damage. Moreover, one can distinguish more than 30 clinical subtypes. For each of these congenital EB types, genes have been identified whose mutations are responsible for the acquisition of the EB phenotype, in particular, PTC/PTC, PTC/GS, Mis/Mis, GS/GS, and G5 in the case of DEB (*COL7A1*) [5,6]. More than 300 monogenic skin diseases with autosomal dominant or recessive forms of inheritance have been described [7,8].

RDEB is one of the most severe and dangerous EB forms that can be characterized as a rare, debilitating autosomal recessive disease caused by biallelic mutations in *COL7A1*, the gene encoding collagen type VII (C7) [9]. Currently, more than 700 mutations have been revealed in the *COL7A1* gene [10]. These mutations are manifested either by the complete absence of C7 (the most severe clinical variants) or by its insignificant production (milder variants). This accurate correlation has been proven between the residual expression of C7 and the milder course of the disease. In addition, the degree of C7 immunopositivity detected by immunofluorescence analysis along with genotype analysis is an important factor in determining the RDEB phenotype [1].

It should be noted that both recessively inherited RDEB and autosomally inherited DDEB are associated with mutations in the *COL7A1* gene encoding C7; however, DDEB causes milder EB manifestations, whereas RDEB is the most destructive [10].

The C7 protein is produced by two types of skin cells: Ks to a greater extent (90–97%) and dermal fibroblasts (DFs) to a lesser extent (3–10%) [11]. C7 is secreted into the extracellular matrix (ECM) of the dermis where it forms anchoring fibrils (AFs) that attach the dermis to the BMZ. C7, as the main component of AFs, plays an essential role in the formation of a stable DEJ [2,9]. RDEB causes a significant reduction in or the termination of C7 synthesis; therefore, there is no formation of functional AFs, which leads to the fragility of skin that is frequently injured with the appearance of multiple, often generalized, large blisters with hemorrhagic content formed in any skin region [12]. The blister surface can be easily damaged, which results in the formation of extensive and deep erosions that are slow-healing and frequently turn into chronic wounds. In many cases, scars are formed on the site of healed wounds, which can cause mitten deformities of the hands and feet leading to the early disability of patients [13,14]. It is important to note that the symptoms of RDEB are not limited only to skin lesions. RDEB is also a systemic disease accompanied by injuries to mucosal linings and organs and an increased risk of developing aggressive forms of squamous cell carcinoma (SCC), which leads to significant mortality at a young age [4,14,15,16]. It has been shown that more than half of RDEB patients acquire SCC by the age of 30. According to the latest report from the Dutch EB Registry, the incidence of RDEB is 5.5 cases per million newborns, and the prevalence is 2.1 cases per million people [17].

The symptoms of RDEB already appear in a child at birth or in the first hours of life. RDEB patients experience constant pain and itching. There are frequent exacerbations accompanied by disturbances to the physical and mental state, which places a heavy burden on both patients and their family members [4,18].

RDEB patients receive no effective therapy due to the lack of RDEB treatment methods taking into account the genetic nature of the disease [19]. Etiopathogenetic treatment is limited to therapeutic palliative measures aimed to relieve a patient’s debilitating symptoms; in addition, RDEB patients also receive psychological and social assistance, take measures to prevent injuries and blistering, protect against infections through skin defects, and receive nutritional support and the treatment of complications such as hand contracture and esophageal dilatation [1,4]. Approaches to RDEB treatment that do not eliminate the root cause of the disease only temporarily alleviate the condition of patients while systemic inflammation continues to develop, and the disease progresses rapidly.

However, the last two decades have been marked by a breakthrough in the knowledge of molecular mechanisms of RDEB, as well as methodological and technological advances, which promoted the significant progress in the treatment of this disease for the nearest future. At the present time, two main directions in the RDEB treatment approaches are considered: (1) the development of genetically reasonable methods targeted at the secondary inflammation-related pathology and, thus, allowing one to reduce inflammation in tissues/organs, thereby reducing the disease severity by slowing down its progression and improving the life quality; and (2) the creation of technologies that could replace or correct the defective *COL7A1* gene and, therefore, to affect the root cause of the disease. The research in these directions is conducted at many scientific and clinical centers around the world [1,20,21,22].

The first direction comprises the cell-based therapy approaches studying the potential use of cultured stem cells and mature cells, both autologous and allogenic [23,24,25,26,27,28,29,30,31,32]. The allogenic cells obtained from various tissues of healthy donors can be used topically for wound healing and systemically to reduce inflammation in the patient’s body. The autologous cells obtained from the tissues of patients and subjected to genetic modification can also be used both locally and systemically.

The second direction includes gene therapy methods since the monogenic etiology of RDEB makes this disease an ideal candidate for the use of gene methods. The aim is to restore the constant production of functional C7 by means of adding, replacing, or modifying the gene encoding C7, i.e., acting at the level of DNA, RNA, proteins, or at the cellular level [33,34]. To date, two methods of gene therapy have become the most widespread, namely (1) gene replacement therapy by replacing the mutant allele of the *COL7A1* gene in skin cells with the exogenous full-sized wild-type cDNA using transduction [35]; and (2) gene-editing therapy. The transduction is implemented using various viral vectors according to two main strategies (ex vivo and in vivo), while gene editing is performed using specific nucleases (for example, TALEN [35,36,37] and CRISPR/Cas [19,20]). Both gene therapy methods can precisely cause the irreversible changes in the gene nucleotide sequence in the region of DNA mutations [38,39].

The purpose of this review is to provide a detailed analysis of the current status of a new generation of biomedical products for gene and cell RDEB therapy, whose development is based on the most advanced methodological and technological achievements in the field of medicine. This review considers both the advantages and disadvantages of the presented products, as well as the prospects for their translation into clinical practice. The significance and timeliness of such a review is associated with the urgent need for both researchers and practitioners to have a complete understanding of the current state of affairs in the development of drugs and strategies that could eliminate the root cause of this severe genetically determined disease. The demand for such knowledge among physicians, especially about products that are already available to patients or are close to being available, is extremely high since there is an understanding that the radical treatment of RDEB patients is possible only through gene and cell therapy.

## 2. The Main Directions in the Development of RDEB Cell Therapy

The results of a number of studies suggest that the strategies based on the use of cells producing the C7 protein and involved in reparative and regenerative processes are the most promising in RDEB cell therapy. We are referring to Ks, dermal fibroblasts (DFs), bone marrow (BM) cells, and mesenchymal stem (stromal) cells (MSCs) [40,41,42,43,44,45,46].

### 2.1. Keratinocytes and Dermal Fibroblasts as the Cellular Components of RDEB Therapy

Keratinocytes are the main cells of the interfollicular epidermis. The progenitors of Ks are basal epidermal stem cells (ESCs) that possess the high mitotic activity maintaining the constant renewal of the epidermis [40]. Two populations of proliferating cells can be distinguished in the epidermis. The first cell population is ESCs that can make up to 10% of the basal layer cells, while the second cell population comprises the keratinocyte-forming transit-amplifying progenitor cells that are the daughter generation of ESCs [47]. These two cell populations make up the pool of basal Ks responsible for maintaining the number of Ks in the epidermis. Epidermal homeostasis is maintained by the balance between the proliferation of basal Ks and their differentiation into Ks of the skin surface layers [48,49]. Ks synthesize a number of important proteins involved in the formation of the DEJ structure, including C7, which is the main protein of anchoring fibrils (AFs) [47], as well as a number of important mitogens (fibroblast growth factor, insulin-like growth factor, epidermal growth factor, endothelin-1) contributing to the proliferation of skin cells and skin repair processes [49,50].

Dermal fibroblasts are located in the dermis, the middle layer of the skin. They have mesenchymal origin, possess both mitotic and biosynthetic activity, and participate in the production and organization of dermal ECMs. In addition, DFs are involved in the stimulation of epidermal morphogenesis and vascular growth as well as in the production of cytokines and growth factors that have autocrine and paracrine effects. Furthermore, DFs possess immunomodulatory and wound healing properties [42,43,44,50]. DFs are easily cultured, retain the diploid karyotype, do not express antigens of the main histocompatibility complex class II (MHC II), do not exhibit oncogenic properties, and maintain their biosynthetic activity after transplantation into the dermis [51,52]. Both allogenic DFs and autologous DFs are available for cell therapy. The autologous DFs (autoDFs) are capable of restoring the ECM [51,52], whereas the allogenic DFs (alloDFs) act in a paracrine manner, producing cytokines and growth factors which stimulate cell proliferation and differentiation along the wound edges, thereby promoting wound healing [42,43,44].

Each type of these cells (Ks and DFs) has a particular area of application. A number of researchers consider Ks as an ideal cellular component for repairing the skin in RDEB patients [53,54,55]. The successful optimization of primary K culture and subsequent ex vivo manipulation of generated ESCs (known as holoclones) served as the basis to develop the transgenic epidermal transplants (TETs) capable of regenerating the skin after grafting [3,47,56].

Both cell types (Ks and DFs) contribute to the synthesis of DEJ components; however, it has been revealed that Ks retain the higher proliferative potential, secrete more C7, and maintain adhesion between the epidermis and the dermis [57] (more information on the use of Ks for the treatment of RDEB patients can be found in the section entitled “The RDEB gene therapy”). According to Jackow J. et al. (2016), the intradermal administration of DFs promotes the production of more sustainable C7 [58].

#### The Application of Dermal Fibroblasts in RDEB Cell Therapy

In addition to the fact that DFs and Ks are the main producers of C7 in the skin, they also possess immunomodulatory and wound healing properties [43,44]. Preclinical studies conducted on C7 hypomorphic mice (animal model of RDEB) have demonstrated that the intradermal administration of human DFs promotes the increased content of C7 and AFs in DEJs and, thus, wound healing [59].

Based on these data, Wong T. et al. (2008) conducted the limited clinical study and showed that the single administration of alloDFs resulted in the increase in *COL7A1* expression and C7 production observed in three out of five RDEB patients for 3 to 9 months, while the stimulation of wound healing was also noted [60] (Table 1). The cell suspension was injected intradermally along the wound edges. The study results confirmed that alloDFs have low immunogenicity and cause no pathological reaction in the patient’s body. The researchers have also revealed that the clinical effect was the most pronounced in patients with the residual baseline C7 production compared to those patients who had this protein be completely absent. The noted effect appears to result from the increase in the level of endogenous C7 in patients with the residual functional activity of the *COL7A1* gene due to the stimulation of its expression in resident DFs and Ks by the transplanted alloDFs. The mechanism of alloDFs’ action can be associated with the paracrine effect of alloDFs, which induces the production of the epithelial growth factor (HB-EGF) in nearby Ks. HB-EGF, in turn, enhances the expression of *COL7A1* gene in resident skin cells, and, thus, increases the production of C7 [61,62]. Wong T. et al. (2008) performed the single intradermal injections of alloDFs to five RDEB patients and noted the increase in C7 and AF content in DEJs after 2 weeks and 3 months following injections [60]. No serious side effects have been reported. According to Wong T. et al., the main effect of alloDFs is the increase in COL7A1 mRNA level in the skin of recipients, which was accompanied by the production of mutant C7 in DEJs and the formation of rudimentary AFs [60]. Apparently, this mutant protein was partially functional and capable of increasing the degree of adhesion in DEJs. It should be noted that the transplanted alloDFs were detected in the dermis for a short time (no more than 2 weeks) [62].

Two randomized placebo-controlled double-blind clinical trials conducted by Petrof G. et al. (2013) [23] and Venugopal S. et al. (2013) [24] using alloDFs have detected re-epithelization in most RDEB patients during the first 28 days, while the obtained result was maintained for 6–12 months. Venugopal S. et al. noted that after administration of alloDFs, the area of large wounds was decreased by 50% compared to similar wounds with placebo-controlled injections (suspension solution) [24]. However, both studies (with five and eleven patients, respectively) have revealed no significant difference between the effects of alloDFs and the suspension solution after 28 days.

The clinical trial conducted by Moravvej H. et al. (2018) with the participation of seven RDEB patients having seven wounds each compared the effects of intradermal injections of alloDFs and DFs seeded on amniotic membrane scaffolds (FAMS)—amniotic membranes of healthy term neonates who were born via caesarean section—using the standard wound care (SWC) with Vaseline^®^ gauze as a control [63]. The clinical trial revealed the significant decrease in wound sizes in the group of patients receiving alloDF injections compared with the group with FAMS injections, while no changes were detected in the control group. The researchers suggested that alloDF injections promote the healing of RDEB wounds and are significantly more effective compared to FAMS or the control treatment.

The other research groups focused on developing methods to treat the RDEB patients’ skin wounds using genetically modified autoDFs. Georgiadis C. et al. (2016) used the self-inactivating (SIN) lentiviral vector (LV) encoding codon-optimized transgene *COL7A1* to transduce autoDFs obtained from RDEB patients [64]. The researchers in vitro and in vivo (human/murine xenograft model) have demonstrated the long-term production of recombinant full-sized C7 and the presence of functional AFs in DEJs. Marinkovich M. et al. (2018) presented the data of a clinical trial phase I/II (NCT02810951) obtained with the participation of five RDEB patients that were treated using autoDFs genetically modified by means of the FCX-007 lentiviral vector (Fibrocell Technologies Inc., Exton, PA, USA) [25]. The linear production of C7, the deposition of AFs, as well as healing of up to 80% of the treated wounds was observed after 3 months following the injections. During 13 months of the follow-up, no complications or side effects were noted. Currently, patients are recruited to conduct the phase III multicenter, intra-patient randomized, controlled, open-label study of FCX-007 for the treatment of chronic wounds (NCT04213261) [1]. The researchers suggested that the use of autoDFs genetically modified with C7 is the effective method to treat skin wounds in RDEB patients, which allows one to improve the skin condition and function, as well as the quality of patients’ life.

The results comparable in safety and initial efficacy were obtained by Lwin S. et al. (2019), who conducted a phase I clinical trial (NCT02493816) with the participation of four adult RDEB patients who received three intradermal injections of autoDFs modified with *COL7A1* in intact skin areas [65]. All four patients tolerated the intradermal injections, while no adverse events and negative immune reactions were noted. The synthesis of full-sized C7 was registered, but no new AFs were revealed. At the same time, the variable results were detected using this therapy method, which may be related to the dose of autoDFs or the skin area of injections. Nevertheless, the researchers consider this method as the promising treatment that promotes wound healing and stimulates the skin functionality of RDEB patients. In addition, the application of this method requires no anesthesia and hospitalization.

**Table 1 ijms-25-10270-t001:** Clinical trials of DFs for RDEB treatment.

Research Group	Number of RDEB Patients	Clinical Trial (N)	Reference
alloDFs			
Wong T. et al. (2008)	5		[59]
Petrof G. et al. (2013)	5	ISRCTN67757229	[22]
Venugopal S. et al. (2013)	11		[23]
Moravvej H. et al. (2018)	7		[62]
autoDFs			
Marinkovich M. et al. (2018)	5	phase I/II (NCT02810951)	[24]
	recruiting patients	phase III (NCT04213261)	[1]
Lwin S. et al. (2019)	4	phase I (NCT02493816)	[64]

### 2.2. The Prospects for Bone Marrow Cell Transplantation in RDEB Treatment

Hematopoietic stem cells of the bone marrow (HSC-BM) are intensively involved in wound healing by providing inflammatory cells that produce cytokines required for this process [66]. In addition, HSC-BM can serve as a source of skin progenitor cells [67,68] and are capable of homing after system injections [69]. These data provided the basis for considering the transplantation of allogenic BM (BMT) cells as the promising strategy for RDEB therapy. The data showed that the purified HSC-BM transplanted into lethally irradiated mice are able to differentiate into epithelial cells [45].

To clarify the role of BM cells in skin wound healing, Badiavas E. et al. (2003) conducted the study using the BM cells labeled with green fluorescent protein (GFP) and transplanted into non-GFP mice [45]. Subsequent histological analysis revealed the significant amount of the transplanted BM cells in the skin and especially in wounds. The researchers suggested that the skin damage stimulates the engraftment of BM cells into the skin and induces their differentiation into non-hematopoietic skin cells, which probably promotes the regeneration of the damaged skin tissues.

The preclinical studies were conducted on RDEB mice using the allogenic BM cells labeled with GFP (mice expressing green fluorescent protein were used as donors of BM cells) [26,69]. The results of BMT confirmed that the BM cells were able to migrate to the wounded skin area and elevate C7 content in DEJ, thereby increasing the lifespan of RDEB mice. Tolar J. et al. (2009) conducted the study to estimate the engraftment of BM cells into the skin after congenic BMT using an RDEB *COL7A1*^−/−^ knockout mouse model [26]. The results showed the increase in the level of *COL7A1* mRNA and in the amount of newly synthesized rudimentary AFs, while the lifespan of mice was increased by 15%. The high survival rate of *COL7A1*^−/−^ mice with the introduced allogenic BM cells was also confirmed by several studies [70,71].

Based on the encouraging results of preclinical studies performed by Wagner J. et al. (2007–2009), the clinical trial (phase I/II, NCT00478244) was conducted with the participation of six RDEB patients using HLA-compatible transplantation of BM cells after myeloablative chemotherapy (myeloablative conditioning) (Table 2) [27]. The healing of chronic wounds and the decrease in blister formation were observed in patients after 30 days following BMT. An analysis of the restored skin areas showed an increase in C7 content in DEJs observed in five out of six patients after 30–130 days, while the formation of new AFs was not detected. The study also showed faster wound healing in patients with initial C7 production, which may be caused by the increase in expression of the *COL7A1* gene due to the paracrine effect of transplanted BM cells. In one patient, C7 production was not observed after BMT; nevertheless, the clinical effect was recorded. This is probably associated with the immunomodulatory effect of transplanted BM cells that reduce inflammation and, therefore, facilitate wound healing. In all RDEB patients, the significant amount of donor cells was revealed in the skin and mucous lining, which confirms homing of the transplanted BM cells into damaged tissues. Analysis of the transplanted cells in the epidermis and the papillary layer of the dermis revealed two main cell populations: hematopoietic CD45+ and non-hematopoietic non-endothelial CD45^−^/CD31^−^. A four-year follow-up confirmed the sustainability of the achieved clinical result.

It is also necessary to take into account such a positive effect of BMT as the development of immune tolerance in RDEB patients to cellular products obtained using the cells of the same donor [72]. Thus, the prospective open-label clinical trial on the transplantation of epidermal grafts (NCT02670837) obtained from the same donor using the CELLUTOMETM Epidermal Harvesting System revealed that eight patients with RDEB (received 35 epidermal allografts in total) demonstrated the longer engraftment of these allografts (at least for 3 years) without any signs of rejection [73].

Nevertheless, besides the positive effect of allogenic BMT, some studies have noted the high risk of mortality. Thus, Wagner J. et al. (2010) observed the life-threatening complications associated with previous myeloablative conditioning in two patients [27]. Later, Geyer M.B. et al. (2015), using the improved protocols of myeloablative conditioning, conducted BMT in two RDEB patients [74]. The study revealed the dermal chimerism and the short-lived clinical effect was observed, but no C7 products were detected.

At the international EB meeting in 2017, Uitto J. et al., after summarizing the results of clinical studies on the use of BMT, reported that not a single patient was cured, but the mortality rates were lowered, and the decrease in blister formation was observed in some patients, as well as an improvement in the life quality [75].

Moreover, one should note the negative experience of using cord blood transplantation (CBT) in the clinical trial initiated by Gostynska K.B. et al. in 2014 with the participation of two RDEB patients. As a result, both patients died due to transplantation-related complications [28]. The study was prematurely closed.

The continuing high risk of mortality and some unanswered questions, in particular, how BMT contributes to improving the skin condition, given that not all patients with a positive effect on the skin had C7 production and AF formation, led to a reduction in the number of clinical trials in this direction [53].

Thus, despite the positive clinical effect of BMT (taking into account the systemic effect on the body, including mucosal linings), there is the high risk of mortality due to the “graft-versus-host” disease and serious adverse events associated with the toxicity of myeloablative conditioning. The method requires serious improvement.

**Table 2 ijms-25-10270-t002:** Clinical trials of alloBM cells for RDEB treatment.

Research Group	Number of RDEB Patients	Clinical Trial (N)	Reference
BM cells			
Wagner J. et al. (2007–2009)	6	phase I/II, NCT00478244	[27]
Geyer M.B. et al. (2015)	2		[74]
CBT			
Gostynska K.B. et al. (2019)	2	prematurely closed	[28]

### 2.3. Current Strategies Using Mesenchymal Stem (Stromal) Cells for RDEB Therapy

The alternative approach to treat RDEB is the use of multipotent mesenchymal stem (stromal) cells (MSCs), which can be obtained from many body tissues: bone marrow [76], the skin [77], gums [78], adipose tissue [79], and umbilical cord blood [80], as well as, from any postnatal tissue [46].

Currently, two main features of MSCs serve as a basis for their isolation and identification. The first feature is the ability to self-renew and undergo differentiation to the tissues of mesodermal origin (multipotent differentiation potential). The second feature is the ability to express the markers that are characteristic of stromal cells (CD105, CD73, and CD90) and the disability to express the markers characteristic of hematopoietic and endothelial cells (CD45, CD34, CD14, CD11b, CD79a, CD19, and HLA-DR) [81,82]. These criteria of MSC isolation and identification play an important role in the development of regenerative medicine methods since the standardization of cellular material is a prerequisite for the successful implementation of biomedical technologies based on the use of MSCs.

The particular interest in MSCs is caused by the fact that these cells have the ability to secrete various growth factors, cytokines, chemokines, and exosomes involved in tissue repair through paracrine action [83], which leads to increased migration, proliferation, and differentiation of endogenous progenitor cells [83,84]. MSCs also promote vasculogenesis and epithelialization, have an antiapoptotic effect by preventing cell death and increasing antiapoptotic cell activity, and contribute to restoring the local microenvironment of stem cells in damaged tissues [85,86]. MSCs possess anti-inflammatory and immunomodulatory properties [87,88,89], inhibiting the oxidative tissue damage through the PGE2-dependent reprogramming of proinflammatory M1 macrophages into anti-inflammatory M2 macrophages. In addition, MSCs demonstrate the increased secretion of anti-inflammatory cytokine IL-10 [90] and the increased release of IL-6 and TGF-β inhibiting neutrophil recruitment by cytokines of endothelial cells [88]. MSCs have demonstrated in vitro the ability to transfer C7 protein and *COL7A1* mRNA to neighboring cells via extracellular vesicles [91]. All these mechanisms associated with MSCs promote the wound healing and reduce the inflammatory processes in the body [83,86,89].

No expression of major histocompatibility complex II (MHC II) was revealed in the cultured MSCs, which excludes the development of immunological adverse reactions after MSC administration [89,92]. At the same time, it has been shown that MSCs can be used both intradermally and systemically, which is especially important in the case of RDEB patients who not only have skin lesions, but also systemic inflammatory diseases [29,30,93], whose pathogenesis is accompanied by proinflammatory cytokine IL-1β constantly produced by Ks that play an essential role [94,95]. Besides the local effect on surrounding cells causing the persistent inflammation of the skin, epidermal cytokine IL-1β being over-released can damage other tissues and organs through systemic circulation and contribute to the development of life-threatening complications of RDEB, such as nephro- and cardiomyopathy [95]. MSCs have immunomodulatory and anti-inflammatory effects associated with the influence of these cells on proinflammatory signaling pathways, including the epidermal cytokine IL-1β (due to the adaptive release of IL-1 receptor antagonist [IL-1RA]) [86,89,92,96]. Therefore, these data confidently evidence that the systemic use of MSCs can contribute to improving the condition of RDEB patients [84,87]. The well-studied subpopulation of MSCs expressing IL-1Ra has been described recently [96]. MSCs are also able to alter the cytokine secretion profile of dendritic cells, which leads to the increase in secretion of cytokine IL-10 possessing the anti-inflammatory effect and the decrease in secretion of IFN-g and IL-12 that have the inflammatory effect [97]. MSCs also have the ability to regulate the immune response by increasing the number of CD4 + CD25 + FoxP3+ regulatory T cells [98].

It was also revealed that the systemically introduced MSCs are able to migrate to the damaged tissue regions due to their homing characteristics [99] and promote their reparation by paracrine action [100]. One should note another anti-inflammatory mechanism that is associated with IL-1RA synthesized by MSCs. In skin wounds, IL-1RA causes reprogramming M1 macrophages into M2 macrophages and, therefore, decreases the infiltration of tissues by M1 macrophages and promotes wound healing [90,100].

MSCs isolated from BM (BM-MSCs) obtained from healthy donors were used for the first time to treat RDEB patients. The results of preclinical studies conducted on immunodeficient mice confirmed the safety of administration of human BM-MSCs both intradermally [93] and intravenously [26], while the healing of skin wounds and improvement in the condition of animals was noted in both cases. Thus, Tolar J. et al. (2009), using a *COL7A1*– mouse model, demonstrated that intravenously injected wild-type congenic BM-MSCs were able to migrate into the skin and mucous linings, increase C7 production and AF formation, and promote wound healing [26].

The obtained data served as the basis for conducting clinical trials focused on the possibility of using BM-MSCs to treat RDEB patients. The researchers made the assumption that the introduced MSCs can migrate to the damaged zones where they reduce systemic inflammation and heal the skin wounds and mucous linings due to the paracrine action through their immunomodulatory, anti-inflammatory, and proangiogenic effects [83,85,86,87,89]. Kosaric N. et al. (2019) demonstrated that the introduced allogenic cells do not remain in the body for a long time, while the improvement in the condition of RDEB patients occurs due to the paracrine effect of the secretome released by transplanted MSCs [101].

El-Daraut M. et al. (2016) reported the results of the randomized double-blind clinical trial with the participation of 14 pediatric RDEB patients who were intravenously injected with allogenic BM-MSCs [30]. The observation period was 12 months. All patients showed an improvement in the clinical picture, the decrease in the number of existing blisters, and the absence of new ones. Histological examination of skin biopsies revealed the increase in the amount of C7 and the formation of new AFs. No side effects or adverse events were observed.

In the clinical study conducted by Petrof G. et al. (2015), 10 children with RDEB were injected intravenously three times with allogenic BM-MSCs [31]. BM-MSC infusion was carried out for 10 min, and after 1 h, the patients returned to their normal lifestyle. The results evidenced the pronounced clinical effect, i.e., wound healing, decreased blister formation, reduced skin erythema, and pain reduction. No side effects have been reported. Patients noted the normalization of sleep and improvement in life quality. The clinical effect persisted for 4–6 months. However, unlike the results obtained by El-Daraut M. et al. (2016), no increase in C7 content and AF formation was detected [30].

In the clinical study (prospective, phase I/II, open-label study) conducted by Rashidghamat E. et al. (2020), nine adult RDEB patients received two intravenous infusions of BM-MSCs obtained from healthy donors [32]. It has been revealed that BM-MSCs were well tolerated and no serious adverse events were observed for 12 months. The transient reduction in disease manifestations was observed as well as the significant reduction in itching. The transient increase in C7 level was detected in one patient. The total number of blisters on the body of RDEB patients was decreased by an average of 2.8 times on day 28 and 2.9 times on day 60.

It should be noted that the increase in C7 content and the formation of new AFs in both children and adults was not always recorded in RDEB patients after BM-MSC infusions, whereas wound healing and the decrease in the number of blisters compared to the baseline level were observed in almost all patients. The researchers suggested that such an effect may result from the increased level of other adhesion proteins located in DEJs, as well as the decreased level of inflammatory mediators (in particular, IL-1β), or the increased level of HMGB-1 in adult RDEB patients, which indirectly leads to the increased adhesion in DEJs [93,102]. According to the researchers, the lower clinical effect detected in adult RDEB patients compared with children can be explained by the fact that, in adults, the disease is associated with the greater and prolonged systemic inflammation and fibrosis accompanied by the significantly higher risk of developing squamous cell carcinoma (SCC). Thus, SCC was developed in two participants of the clinical study conducted by Rashidghamat E. et al. (2020) after 6 and 7 months, respectively, following BM-MSC infusions [32]. However, the researchers do not directly link these events and believe that conclusions should be drawn after conducting large-scale clinical trials since RDEB patients initially have a significantly increased risk of developing SCC due to the chronic inflammation of skin wounds and fibrosis [15]. To date, more than 700 clinical studies have been conducted using MSCs from various tissue sources to treat different diseases, and none of them have registered the development of malignant neoplasms. However, one of the key issues in future clinical studies should be the question whether the infusion of MSCs can cause or accelerate the development of SCC in RDEB patients [32].

In general, the systemic administration of BM-MSCs is considered to be a promising method of treatment that is well tolerated by patients.

The intradermal administration of BM-MSCs is also the promising approach to treat skin wounds in RDEB patients. The preclinical studies using a xenograft model demonstrated that the intradermal administration of BM-MSCs promote the healing of skin wounds, increase C7 production, and stimulate AF formation [93,102]. The placebo-controlled clinical trials (RDEB; OMIM 226600) were conducted by Conget P. et al. (2010) with the participation of two RDEB patients who were intradermally injected with BM-MSCs along the edges of chronic wounds [29]. A week after the cell injections, C7 was detected along the BMZ, while the continuous DEJ structure was observed. Re-epithelization of chronic wounds was observed only at the sites of MSC injections. In both patients, the achieved clinical effect persisted for 4 months. The researchers concluded that the intradermal administration of allogenic MSCs may lead to increased C7 content in DEJs, prevent blistering, and improve wound healing in RDBE patients.

According to Kuhl T. et al. (2015), the intradermal administration of BM-MSCs cause a clinical effect similar to that of of DFs; however, it requires 10 times less cells [102]. Using the fewer cell amount can allow one to reduce the volume of an injection, thereby reducing the pain associated with intradermal injections and treating the larger areas of the body surface. Another advantage of MSCs compared to DFs is the greater immunomodulatory effect leading to the greater reduction in inflammation in skin wounds [103]. Thus, one can conclude that BM-MSCs compared with DFs have a greater potential for wound healing, restoring skin integrity, and reducing the severity of clinical manifestations in RDEB patients.

Another source of MSCs having a comparable clinical effect to BM-MSCs is human umbilical cord blood. Lee S. et al. (2021) conducted the first clinical trial (phase I/IIa, ClinicalTrials.gov NCT04520022) to evaluate the safety and clinical efficacy of intravenous infusions of allogenic MSCs obtained from umbilical cord blood (hUCB-MSCs) [104]. Six patients with RDEB, including four adults and two children, received three intravenous infusions of hUCB-MSCs with an interval of 2 weeks between them. The patients were observed for 8–24 months after therapy. The study showed that the treatment was not accompanied by any serious side effects. The decrease in severity of RDEB according to the Birmingham index was registered, as well as the reduction in lesions of the body surface area, the decrease in the number of blisters, reduction in pain and itching, and the improvement in life quality, while the maximal effect was observed after 56 to 112 days following therapy. The injections of hUCB-MSCs also induced the reprogramming of macrophage phenotype M1 to macrophage phenotype M2, which decreased the infiltration of skin wounds by M1 macrophages and mast cells. The increase in C7 production in DEJs was observed in one out of six patients on day 56. Thus, the study results showed the safety and clinical efficacy of intravenous administration of hUCB-MSCs in RDEB patients. Over time, the clinical effect of hUCB-MSCs gradually weakened, but the steady improvement was observed in some patients up to 6 months.

It has been revealed that in the skin of RDEB patients, the density of nerve fibers is decreased, whereas the number of activated mast cells is increased, which appear to cause neuropathic pain and itching observed in RDEB patients [105,106]. Studies have shown that the systemic use of hUCB-MSCs decreases the amount of neuropeptides produced by sensory cells and reduces the levels of substance P and other factors involved in neurogenic inflammation, neuropathic pain, and itching [107,108]. This is also facilitated by the significant reduction in the infiltration of skin wounds by mast cells, which are actively involved in inflammatory processes in the skin of RDEB patients. The results obtained by Lee S. et al. (2021) [103] using hUCB-MSCs are comparable with the results of clinical studies on the systemic administration of BM-MSCs [30,31,32]. However, the umbilical cord blood is the more convenient source of MSCs due to the noninvasiveness of its collection procedure and the rapid availability from cord blood banks [109]. It has also been shown that hUCB-MSCs have a higher proliferative capacity and a lower immunogenicity compared with BM-MSCs, as well as greater immunosuppressive and regenerative potentials compared with BM-MSCs [110,111].

The promising treatment strategy for RDEB patients has been proposed by Liao Y. et al. (2018), i.e., the use of stem cells isolated from the placenta (SCsP) by means of technology developed by Celgene Cellular Therapeutics (CCT processing) [112]. It has been shown that SCsP contain a high percentage of non-hematopoietic (MSCs) and hematopoietic stem cells (HSCs) since the placenta is not only the most important system for fetal survival and growth, but also serves as the hematopoietic organ during intrauterine development [113]. SCsP possess high regenerative properties [112], do not express MHC II, and can be considered as universal cells for allogenic transplantation in RDEB patients. The results of preclinical studies conducted on *COL7a12/2* mice demonstrated the safety and efficacy of methods using SCsP. The intravenous injections of SCsP resulted in the sufficiently high level of C7 production in the DEJ region and improved wound healing. In these mice, C7 is completely absent; therefore, all the registered amount of C7 is produced by transplanted SCsP [113]. The mechanism of C7 production and skin regeneration after the use of SCsP is currently actively studied. Undoubtedly, this mechanism has the multifactorial nature comprising both HSCs and MSCs, which are parts of SCsP [113].

Recently, the promising subpopulation of immunomodulating MSCs called ABCB5+ MSCs has been identified in the human dermis [114,115]. The new subpopulation of MSCs expresses the classical MSC markers [81,82], reliably maintains the capacity of self-renewal and tri-lineage differentiation in vitro, and possesses a suppressive effect on effector T cells and an enhancing effect on regulatory T cells [114]. Beken S. et al. (2019) has demonstrated that the subpopulation of MSCs can be successfully isolated from the human dermis with high purity using the single marker (P-glycoprotein ABCB5), and used as the well-characterized homogeneous dermal MSC line [116]. This finding is significant for regenerative medicine since the isolation of MSCs by the single marker allows one to standardize the cellular preparations and obtain statistically reliable results in clinical trials [117,118,119]. The subpopulation ABCB5 + MSCs is characterized by the pronounced immunomodulatory and anti-inflammatory effects, including the effect on the macrophage population, thereby decreasing the infiltration of wounds by M1 macrophages [116] and reducing the levels of neutrophils [120] and regulatory T cells [114]. At the same time, ABCB5 + MSCs improve the blood supply of tissues producing endothelial vascular growth factor (VEGF) and enhance angiogenesis in the damaged skin tissue.

It has been revealed that ABCB5 + MSCs are the excellent source of the key cytokine IL-1Ra that effectively suppresses the inflammation associated with M1 macrophages causing various disorders of chronic wound repair [116,118,121]. Beken S. et al. (2019) using the intradermal injections of ABCB5 + MSCs in a mouse model for the first time detected the synthesis of IL-1Ra that effectively suppressed the significant inflammation in wounds, which directly correlated with their healing [116]. The researchers have demonstrated that ABCB5 + MSCs injected along the edges of chronic wounds release cytokine IL-1RA, which transforms the extremely active M1 macrophages into M2 macrophages, promoting wound healing.

It has been proven that chronic wounds, regardless of their etiology, are characterized by the consistently high number of extremely activated M1 macrophages [122]. M1 macrophages possess the increased secretion of proinflammatory cytokines such as IL-1β and TNFα, which, along with proteases and reactive oxygen species, lead to damaged skin tissues and DF senescence, thereby preventing wound healing. In contrast to normal wound healing, the chronic course is accompanied by the destroyed reprogramming of M1 macrophages into M2 macrophages that produce growth factors and metabolites stimulating tissue repair [123]. On the contrary, the cytokines IL-1β and TNFα secreted by M1 macrophages maintain permanent autocrine activation of M1 macrophages forming a vicious circle, contributing to persistent inflammation that prevents wound healing [116]. The paracrine mechanisms initiated by ABCB5 + MSCs being introduced to the wounds infiltrated by M1 macrophages break this vicious circle due to the adaptive secretion of IL-1RA and cause the replacement of M1 macrophages by M2 macrophages, thereby promoting wound healing [116,121].

The preclinical studies conducted on the mouse RDEB model using single and multiple intravenous injections of ABCB5 + MSCs demonstrated the safety and clinical efficacy of this method, which resulted in an improvement in the clinical condition, the prolonged life of animals, and the healing of skin wounds [124].

Based on the results of preclinical studies, clinical studies have been conducted, which also demonstrated that ABCB5 + MSCs can reduce the RDEB severity and heal chronic wounds [119,125,126,127]. Thus, Kiritsi D. et al. (2021) conducted the international single-arm clinical trial with intravenous infusions of ABCB5+ MSCs to RDEB patients (phases I/IIa Clinicaltrials.gov NCT03529877; EudraCT 2018-001009-98). Three intravenous infusions of ABCB5+ MSCs were performed on 16 patients (aged 6–36 years) [126]. The study results showed the significant reduction in rating scores (Epidermolysis Bullosa Disease Activity, Scarring Index activity, iscorEB clinical disease severity, and itch numeric rating score). No serious side effects or adverse events were observed during the entire follow-up period (12 months).

Dieter K. et al. (2023) conducted a post hoc analysis of this treatment approach to assess its effect on skin wound healing in RDEB patients [127]. The research group used the documentary photographs of the affected body regions for 12 weeks and tracked the development of new wounds. The study results showed that within 12 weeks, the wound healing rate was 76% and the median rate of newly developed wounds decreased by 79%. The researchers concluded that the systemic use of ABCB5+ MSCs promotes the healing of chronic wounds and prevents the formation of blisters and new wounds in RDEB patients.

The results of preclinical and clinical studies allow one to conclude that the specific subpopulation of immunomodulating ABCB5+ MSCs isolated from the dermis of healthy donors is quite promising to treat RDEB. It should be particularly noted that ABCB5+ MSCs can be provided as ready-to-sell medication and considered as an advanced-therapy medicinal product (ATMP) based on the highly purified cell population with the standardized anti-inflammatory therapeutic action (IL-1RA secretion) [118,127].

Thus, the therapy methods based on MSCs represent the valuable strategy to treat RDEB patients using both intradermal and intravenous injections. MSCs are available and easily propagated under conditions specified by GMP. MSCs are already used to treat the number of diseases in humans, which may minimize the safety problems and facilitate the rapid introduction of these cells into clinical practice to treat RDEB patients. However, longer follow-up studies are required to confirm the efficiency and safety of the obtained results and to exclude the risk of developing SCC [126].

To summarize, one should note the sufficiently wide range of cellular products being developed with the suggested high effectiveness in RDEB therapy (Table 3). Some of them have great prospects of entering the medical market in the foreseeable future. In particular, such products include BM-MSCs having high anti-inflammatory, immunomodulating, and regenerative effects, as well as ABCB5 + MSCs isolated from the skin, which demonstrated an even more pronounced clinical effect compared to BM-MSCs. It is extremely important that MSCs can be used both intradermally and intravenously to act systemically on the body, taking into account the nature of RDEB associated with systemic inflammation. The therapy can significantly reduce the disease symptoms and improve the life quality of RDEB patients by reducing inflammatory processes in the body and relieving itching and pain that are persistent and highly disturbing. Kiritsi D. et al. (2021) suggested that the prolonged systemic use of ABCB5+ MSCs may improve the structural integrity of the skin and mucous linings due to the accumulation of C7 in the DEJ region owing to the high adhesion of these cells to damaged tissues (compared to BM-MSCs) and the ability to secrete C7 [126].

Special attention should be paid to FCX-007, the medical product developed on the basis of genetically modified autoDFs (Fibrocell Technologies, Inc., Exton, PA, USA, NCT04213261) that is currently in phase III of clinical trials [65]. In the previous phases of clinical trials, FCX-007, which was introduced intradermally, demonstrated a high wound healing effect [65]. It is important to note that autoDFs isolated from the patients were used in the clinical trials; therefore, one should expect a long-term clinical effect. Given the systemic nature of the disease, one can suggest that the use of FCX-007 in combination with intravenous injections of MSCs with the confirmed anti-inflammatory clinical effect could be particularly efficient.

## 3. Prospects for the RDEB Gene Therapy

The molecular pathology of RBDE is associated with bi-allelic loss-of-function variants of the *COL7A1* gene. In this regard, one of the main approaches to gene therapy of this disease is gene replacement using the full-length *COL7A1* cDNA [53]. At the genomic DNA level, most mutations in *COL7A1* are associated with single-nucleotide substitutions; therefore, the gene editing procedures and other methods of allele silencing or skipping the exons containing pathogenic mutations are considered as the second approach. To date, the most common procedures used for RDEB gene therapy are the gene replacement therapy by means of viral vectors and the gene editing therapy using the programmable nucleases.

### 3.1. Gene Replacement Therapy: Gene Products and Delivery Vectors

The gene replacement therapy allows one to substitute the mutated gene allele for the exogenous full-length wild-type cDNA in the skin cells using the transduction technique. The technology consists of the following steps: (1) Ks (and/or DFs) are isolated from the patient’s skin biopsy; (2) the obtained cells are cultivated; (3) the cell cultures undergo genetic manipulations to obtain the new population of autologous cells possessing the corrected genes encoding the functional COL7A1; and (4) the gene-corrected cells are returned to the patient as the epidermal graft (Ks) or the skin-equivalent graft (Ks and DFs), or intradermally (DFs) [33].

#### 3.1.1. Vectors for Gene Product Delivery

Currently, retroviruses (RVs) (γ-RVs are the most preferred ones), lentiviruses (LVs), and herpes viruses (HSVs) are commonly used as vectors to deliver the gene products to the cell [56]. Retroviruses (RVs) are RNA viruses that infect only mitotically active cells. The main disadvantage of their use is the risk of insertional mutagenesis that can appear when these viruses are integrated into the actively transcribed genes, which can disturb the physiological gene expression [128,129,130]. Nevertheless, the preclinical and clinical studies have detected no insertional oncogenesis in RDEB patients after the γRV-based ex vivo gene therapy [56]. In addition, the self-inactivating γRVs (SIN-γRVs) with a deletion in the U3 region of 30LTR are currently used, which eliminates the strong viral enhancer and, thus, reduces the γRV tendency to activate genes and ensures safer use.

Unlike RVs, lentiviruses (LVs), which are also RNA viruses, infect both the dividing and quiescent cells; therefore, these viruses can transduce the wider pool of cells [131]. LVs have a slightly safer integration profile compared to γRVs [132], while LVs and γRVs have similar LTR-enhancer activity [133]. At the same time, preclinical studies have shown that almost all LV-transduced Ks produced the rearranged, truncated, or fragmented C7 in large amounts [56]. On the contrary, only 25% of RV-transduced Ks contained a few aberrant C7 full-length proteins, which, according to De Rosa L. et al. (2020), allows one to use γRVs to obtain the stable certified packaging of these cell lines that can be used for years. At the same time, it has been proven that LVs are more efficient and less toxic when transducing human primary DFs [64,65]. Currently, lentiviruses are predominantly used to transduce DFs.

Herpes simplex viruses (HSVs) are DNA viruses. HSVs possess the high packaging capacity and the ability to transduce both dividing and quiescent cells. HSVs have a higher safety profile compared to LVs and γRVs. All this makes HSVs the most suitable for in vivo gene therapy procedures [56,134]. The disadvantage is that they remain episomally recircularized in the nucleus, which can lead to a decrease in the duration of the therapeutic effect.

#### 3.1.2. Development of Products for Ex Vivo Gene Replacement Therapy

The following trends are the most important to develop innovative gene replacement therapy products: (1) the creation of transgenic epidermal transplants (TETs) containing Ks; (2) the development of transgenic transplants containing Ks and DFs; (3) the search for procedures to stabilize the number of Ks and DFs in autologous skin equivalents in RDEB patients; (4) the prolongation of the effective functioning of transplants; and (5) the creation of topical products for gene replacement therapy.

##### The Use of Transgenic Epidermal Transplants (TETs) Containing Ks

Both the preclinical studies and the first clinical trials have been conducted with the use of transgenic epidermal transplants (TETs) containing Ks. The encouraging results have been obtained from the preclinical studies on immunodeficient mice whose skin wounds were treated with the grafts obtained on the basis of transduced Ks, i.e., the functional correction was observed, including C7 production, AF formation, and DEJ stabilization for several months after engraftment [53,54,55,57]. Based on these data, the pilot clinical studies were conducted with the participation of RDEB patients whose wounds were covered with TETs containing the genetically modified autologous Ks. The transplants were completely integrated into the skin with a normal functioning of the cells that synthesized C7, not differing from the protein of healthy skin. In addition, the AF formation was detected [34,135,136].

Siprashvili Z. et al. (2016) have conducted a phase I clinical trial (NCT01263379) to test a product later called EB-101, using the γRV delivery vector containing a full-size cDNA copy of the *COL7A1* gene (Table 4) [54]. The researchers restored the production of full-sized C7 in autologous Ks of RDEB patients and performed transplantation of the transgenic epidermal sheets on four patients (six TETs to each patient). After the transplantation, C7 production and AF formation were detected in the skin of patients, the skin integrity was restored, and no serious side effects and adverse events were noted. The linear increase in C7 level was detected in 90% of biopsies after 3 months and 42% of biopsies after 12 months, while AF formation was observed in 71% of biopsies after 3 months and 25% of biopsies after 12 months. The increase in wound healing by 75% or more compared to the baseline was observed in 87% of cases after 3 months and 50% of cases after 12 months (Table 5). According to the researchers, the observed decrease with time is probably associated with the low number of ESCs that can be isolated from biopsies of RDBE patients, as well as with the age-related decrease in the proliferative potential of Ks in culture [54].

Eichstadt S. et al. (2019) have conducted a clinical trial (phase I/IIa, NCT01263379), performing the transplantation of autologous TETs onto the surface of chronic wounds in seven patients with RDEB [136]. After 6 months, an increase in wound healing by more than 50% was observed in 95% of the treated areas, and after 3 years, up to 71% of the treated areas.

In the follow-up study of the product EB-101 (NCT01263379) conducted by So J. et al. (2022), the same seven patients were observed to collect the long-term data (a mean of 5.9 years) on the efficacy and safety of the product [55]. The study results showed that during the fifth year of observation, the healing of treated wounds increased by at least 50% compared to the baseline observed in 70% of cases. No serious side effects or adverse events have been reported.

At the present time, a phase III open-label clinical trial (NCT04227106) using the product EB101 developed by Abeona Therapeutics Inc. [53] to treat RDEB has been accomplished (the study results have not been published yet). EB-101 has been designated as a fast track, regenerative medicine advanced therapy, and rare pediatric disease by the Food and Drug Administration (FDA) and as an orphan drug by both the FDA and the European Medical Agency (EMA).

In addition to the obtained positive results and the proven absence of serious side effects, one should note that over time, the clinical effect of TETs decreases. This may occur for several reasons, one of which is the insufficient amount of ESCs in the primary cell cultures obtained from RDEB patients [137,138]. Another reason may be the significant variability in de novo AF production, ensuring the firm connection between the epidermis and the dermis.

#### Development of the Transgenic Transplants Containing Ks and DFs

Some researchers proceeded from the fact that both types of skin cells (Ks and DFs) associated with C7 production participate in the formation of AFs with typical structure [139,140]. Thus, several preclinical studies have shown that after transplantation—the skin equivalents composed of *COL7A1* that corrected Ks and DFs in immunodeficient mice—one can observe the significant and long-term de novo C7 production [57,141]. The study results obtained in vitro and on the NIH-III-nude strains of immunodeficient mice using the skin equivalents containing Ks and DFs isolated from RDEB patients and from healthy donors confirmed that the expression of the *COL7A1* gene by both cell types involved in the organization of DEJs is required to form the optimally functioning AFs [139,140].

These data were also confirmed by Boyce S. et al. (2006), who treated 40 burn patients with the cultured skin substitutes (CSSs) consisting of autologous Ks and DFs attached to collagen-based sponges [141]. It has been shown that CSSs promote stable wound healing in burn patients with minimal need for skin transplantation and minor scarring formation. According to the researchers, the two-layer graft containing autologous Ks and DFs contributes to paracrine interactions between these cells. As a result, the functional C7 and AFs are formed, promoting wound healing [140,141,142]. The significant role of paracrine interactions between Ks and DFs for optimal skin morphogenesis has also been demonstrated by Wojtowicz A. et al. (2014) [140] and Ghalbzouri A. et al. (2004) [143]. The researchers have shown that the two-layer autologous grafts containing both Ks and DFs are optimal to treat the skin wounds in RDEB patients.

Titeux M. et al. (2010) used the genetically corrected Ks and Ds obtained from RDEB patients to generate the human skin equivalents that were transplanted to immunodeficient mice [144]. The researchers developed the new minimal SIN-gRV containing the *COL7A1* complementary DNA (cDNA) under the control of human elongation factor-1alpha (EF-1alpha) or *COL7A1* promoters. The study results demonstrated the long-term production of recombinant C7, dermal–epidermal adhesion, and AF formation, which confirms the functional viability of the developed skin equivalents.

Taking into account the encouraging results of clinical trials, the project funded by the European Commission and called the GENEGRAFT project has been launched to develop a safe and efficient gene therapy to treat RDEB [145,146]. As part of the project, a phase I/II clinical trial with a participation of 30 RDEB patients is currently being conducted to test the possibility of using the bioengineered two-layer grafts for transplantation into RDEB patients (EBGraft, NCT01874769) (Table 4). EBGraft is the skin equivalent based on genetically modified ex vivo autologous Ks and DFs using the SIN-gRV vector encoding the full-sized *COL7A1*.

**Table 4 ijms-25-10270-t004:** Clinical trials of transgenic transplants for RDEB treatment.

Research Group	Number of RDEB Patients	Clinical Trial (N)	Reference
Single-layer skin transplants (TETs)			
Siprashvili Z. et al. (2016)	4	phase I, EB-101, NCT01263379	[54]
Eichstadt S. et al. (2019)	7	phase I/IIa, NCT01263379	[136]
So J. et al. (2022)	7	EB-101, NCT01263379	[55]
Abeona Therapeutics Inc.		phase III is finished, open-labeled EB-101, NCT04227106	[53]
Two-layer skin transplants			
Gaucher S. et al. (2020)	30	phase I/II EBGraft, NCT01874769 GENEGRAFT project	[145]

##### The Problem of Stabilizing the Number of Ks in Autologous Transplants of RDEB Patients: Possible Solutions

It has been suggested that to maintain the stable number of Ks in autologous skin equivalents in RDEB patients, it is first necessary to increase the number of ESCs in primary cultures [137,138]. To achieve this goal, the induced pluripotent stem cells (iPSCs) were obtained from somatic skin cells (Ks or DFs) of RDEB patients by reprogramming them into the embryonic state using a combination of transcription factors [147,148,149]. The important advantage of iPSCs is their unlimited proliferative potential and pluripotency that allows them to differentiate into different cell lines [148,149,150,151,152], which can be used for genetic modification and subsequent clinical use.

It is interesting that in some studies, the revertant keratinocytes obtained from RDEB patients (revertant mosaicism appears in the normal skin areas in some RDEB patients due to the spontaneous correction of hereditary mutations by subsequent mutations in somatic cells) were used to obtain iPSCs [153]. It has been shown that revertant Ks can be converted into iPSCs that subsequently differentiate into Ks possessing the naturally corrected gene [154,155,156]. Gostynski A. et al. (2014) using an animal model showed that such Ks obtained using this approach can be effectively used for the ex vivo generation of autologous epidermal grafts [157].

Jackowa J. et al. (2019) obtained iPSCs from DFs of RDEB patients using the electroporation of integration-free episomal vectors encoding four reprogramming factors (OCT4, SOX2, KLF4, and C-MYC) [149]. After the correction of the *COL7A1* gene, the iPSCs obtained were differentiated into ESCs and DFs, which served as the basis to create the bilayer skin equivalents transplanted into immunodeficient mice. The study results have demonstrated that the use of genetically modified transplants based on Ks and DFs derived from iPSCs bearing the corrected gene promotes C7 production, AF formation, and dermal–epidermal adherence in the xenograft model. The study confirmed the feasibility of the iPSC-based gene correction technology to treat RDEB patients.

The products developed on the basis of iPSCs are currently at the step of preclinical studies; however, the study results already allow one to suggest that the two-layer transplants based on Ks and DFs derived from iPSCs can serve as clinically effective skin equivalents with prolonged action.

Although ex vivo gene replacement strategies are currently considered as the most advanced according to preclinical and clinical studies, they have certain disadvantages, in particular, one should mention the invasiveness of the skin biopsy procedures (from one to several) to obtain ESCs and traumatic manipulations to prepare the wound bed, which is important for successful engraftment [34,56]. There is also the risk of developing skin wound infections and the appearance of insertion mutagenesis when the gene product (transgene) can be integrated into the cell genome, which disrupts the gene expression and can lead to carcinogenesis [158]. In addition, there is the risk of transgene overexpression accompanied by shutdown of the integrated gene [159], as well as the risk of developing autoimmune epidermolysis bullosa aquisita in patients with null C7 expression [160].

#### 3.1.3. Current Products for in Vivo Gene Replacement Therapy

Beremagene geperpavec (B-VEC) is the topical gene product aimed to restore the C7 synthesis by both types of the patient’s skin cells. B-VEC is currently considered as one of the most successful developments for in vivo gene replacement therapy [161,162]. The distinctive feature of the gene product is that it was engineered on the basis of non-replicating *COL7A1*-containing herpes simplex virus type 1 vector (episomal, non-integrating, eliminating insertional oncogenesis risk).

B-VEC is used as a gel, which does not require hospitalization, anesthesia, and invasive surgical interventions [162]. The functional form of the *COL7A1* gene is delivered directly to the skin cells, then enters the nucleus, and afterwards, the transcripts of the *COL7A1* gene are generated, which allows the cells to produce and secrete the functional C7 protein forming AFs. Since the used vector HSV1 is not capable of replication and insertion into the cell genome, its repeated use is possible. Gurevich I. et al. (2022) have conducted a randomized placebo-controlled phase I/II clinical trial (NCT03536143) [162]. The study results have demonstrated that after the treatment of patients with B-VEC, one can observe increased C7 production, AF formation, and wound surface area reduction. Guide S. et al. (2022) have conducted a phase III double-blind, intra-patient randomized, placebo-controlled trial (NCT04917874) [161]. The study results showed that after 6 months, complete wound healing occurred in 67% of the wounds treated with B-VEC as compared with 22% of the wounds exposed to placebo. In 2023, B-VEC received the full FDA licensing approval for clinical use in the United States [53]. Thus, B-VEC represents the first gene therapy treatment of RDEB (and DDEB) available to patients (Table 5).

It should be noted that when using B-VEC as well as the other gene products, patients with zero C7 production have the high risk of developing adverse immune reactions against the synthesized functional protein; therefore, this therapy is suitable only for patients with residual C7 production and lack of antibodies against the protein product of therapy [160,163].

Thus, it can be concluded that great success has been achieved in gene substitution therapy (both ex vivo and in vivo) of this severe disease due to the significant advances in gene technologies, understanding of the molecular mechanisms of RDEB, and optimization of the protocols for obtaining/maintaining cell cultures.

Therefore, among all gene technologies, the ex vivo gene replacement strategies have advanced the furthest in clinical trials. Taking into account the ability of RV vectors to integrate into the cell genome (which can lead to stable and uncontrollable expression of transgenes), the new safer-to-use viral vectors such as SIN γRV and LV vectors have been developed. It has also been revealed that the optimal formation of structurally normal AFs requires both Ks and DFs to participate in C7 production, since ECM and the paracrine effect play an important role in AF formation [139]. In this regard, special attention is currently paid to the GENEGRAFT project developing the two-layer skin equivalent of EBGraft (NCT04186650) based on the use of autologous Ks and DFs that were obtained from the patient’s skin and transduced ex vivo by full-sized *COL7A1* cDNA using vector SIN γRVs [145,146].

However, the gene replacement therapy has a number of disadvantages, in particular, the ex vivo approach is characterized by invasiveness (repeated skin biopsies are required for successful ESC isolation), as well as serious rehabilitation of the wound bed is required for successful engraftment, and the use of systemic anesthesia (Table 5). For this reason, much attention is paid to the strategies that allow one to avoid these disadvantages. In vivo gene replacement strategies are considered as one of the options, while significant success has been achieved in this direction. Scientists developed and introduced into clinical practice B-VEC, the unique gene product based on the HSV-1 vector that delivers the *COL7A1* gene directly to both types of C7-producing skin cells. Since the HSV-1 vector used is not capable of replication and incorporation into the cell genome, it is possible to reuse B-VEC starting from early childhood. Local application of the drug to wounds and blisters in gel form certainly eliminates the disadvantages characteristic of ex vivo strategies [54,56].

At the present time, in addition to the development of topical products in vivo, great attention is paid to the development of in vivo gene replacement strategies for systemic use given the systemic nature of RDEB.

One should especially emphasize the problem that remains relevant for all genetic engineering strategies, i.e., the high risk of development of autoimmune epidermolysis bullosa aquisita in patients with null C7 expression, which is caused by circulating neutralizing autoantibodies against C7 [160,163]. The results of RDEB gene therapy studies have shown that the presence of even low amounts of antibodies against C7 in RDEB patients obliges to consider such patients with caution as candidates for C7 replacement therapy. In this regard, much attention is paid to the development of approaches that would eliminate the risk of developing autoimmune reactions in RDEB patients when using gene therapy.

Thus, the main tasks in the field of gene replacement RDEB therapy continue to be the improvement in ex vivo technologies, the further development of in vivo strategies, both topical and systemic, and the approaches preventing the development of immune reactions.

### 3.2. The Capabilities of Gene Editing Systems in RDEB Therapy

An alternative approach to restore C7 without the use of full-sized *COL7A1* cDNA transduction is gene editing by means of the specific nucleases that can be targeted to the irreversibly changed gene nucleotide sequence in the DNA-mutated region [3]. Gene editing tools are able to directly correct a mutation in the corresponding gene to restore the gene expression with no risk of developing the insertion mutagenesis that can be observed after the gene replacement therapy [36].

The following synthetic nucleases are used for genome editing: zinc finger nucleases (ZFNs), transcription activator-like effector nucleases (TALENs), and the Clustered Regularly Interspaced Short Palindromic Repeats/Crispr-Associated protein (CRISPR/Cas) system. TALENs [35,36,37] and CRISPR/Cas [38,39] technologies are mainly used for gene editing to treat RDEB.

The action of synthetic nucleases is associated with the formation of nuclease-induced double-stranded DNA breaks (DSBs) to make insertions or deletions in the gene nucleotide sequence, which are restored using mainly two different mechanisms of cellular DNA repair: non-homologous end joining (NHEJ) and using the DNA donor repair template when DSBs are restored by homology directed repair (HDR), which provides the more accurate correction [33,164].

Osborn M. et al. (2013) first achieved up to 2% of genetic correction by means of the TALEN-mediated HDR [130] using DFs obtained from RDEB patients, which were subsequently reprogrammed into iPSCs. The study results showed normal C7 expression and deposition using the teratoma-based skin model in vivo.

Chamorro C. et al. (2016) [165] and Mencía Á. et al. (2018) [166] performed the TALEN-mediated genome editing using Ks and ESCs obtained from RDEB patients. The researchers observed C7 deposition in BMZ and AF formation after transplantation of the grafts obtained using these genetically modified cells to immunodeficient mice.

Aushev M. et al. (2017) using this gene editing system isolated K clones bearing the normal *COL7A1* gene from the skin of RDEB patients [167]. According to the researchers, the obtained K clones can be propagated and used as epidermal layers to treat the skin wounds in RDEB patients.

Preclinical studies have shown that a TALEN is a sufficiently effective editing system capable of specifically recognizing any DNA sequence; however, the significant disadvantage of TALEN is labor consumption and the complexity of cloning [167]. Thus, when the DNA sequence targeted for editing requires to be changed, it is necessary to completely change the TALEN design and clone de novo the entire chimeric TALEN molecule.

In contrast to the TALEN system based on functional proteins, the more modern CRISPR/Cas9 system is directed by non-coding RNA [168]. CRISPR are short palindromic repeats regularly arranged in groups and separated by unique sequences, while Cas9 is the DNA-cutting nuclease [169,170]. CRISPR/Cas9 belongs to systems whose CRISPR regions are transcribed, processed, and form guide RNA in combination with trans-activated CRISPR RNA (tracrRNA). The guide RNA recognizes the target DNA through complementary interactions with the target DNA fragment. After binding to DNA, the guide RNA interacts with the Cas9 nuclease and activate it, causing the rearrangement of its conformation, which results in DNA cutting. CRISPR/Cas9 can be targeted at any DNA sequence in the genome while preserving endogenous promoters for gene expression.

To date, CRISPR/Cas9 was used to edit RDEB DFs, Ks, and iPSCs, which were successfully grafted to RDEB mice with the functional correction of mutations in the *COL7A1* gene [149,171,172,173,174]. Thus, Jackow J. et al. (2019) used the CRISPR/Cas9 (Cas9-gRNAs) system to edit iPSCs obtained from somatic skin cells of RDEB patients [151]. Following the correction of the *COL7A1* gene in iPSCs through HDR, the researchers achieved the successful differentiation of the genetically modified iPSCs into Ks and DFs. It has been shown that a colony (clone) can be formed from one DF progenitor cell and this colony can be isolated/sequenced, which enables the obtainment of a DF clone with normal C7 from one previously edited progenitor cell to produce ex vivo gene products [35].

Webber B. et al. (2016) applied a similar gene editing strategy through the CRISPR/Cas9 system using the primary culture of DFs from RDEB patients to obtain iPSCs, which were differentiated into Ks, MSCs, and hematopoietic progenitor cells [171]. The researchers concluded that the CRISPR/Cas9 system in combination with iPSC technology can be successfully used to create the autologous platform as the basis to produce several types of genetically modified cells, which can be used to treat RDEB patients. Itoh M. et al. (2011) used the analogous method to edit the iPSC genome using CRISPR/Cas9 [151].

Wu W. et al. (2017) used the CRISPR/Cas system (Cas9-gRNAs) for the local delivery of Cas9/sgRNA ribonucleoproteins into the skin of postnatal mice. This system effectively removed the exon with point mutation in the *COL7A1* gene in ESCs of RDEB mice and, thus, restored the correct localization of C7 in vivo [174]. The study revealed the significant decrease in the formation of blisters. The study also demonstrated that gene therapy methods based on the Cas9/sgRNA system are able to correct the defective genes at the level of stem cells in vivo, which provides the long-term (possibly lifelong) restoration of the defective genes. The Cas9/sgRNA system also allows one to avoid the risk associated with viral delivery systems (e.g., the possible integration of viral DNA into the genome and activation of virus-induced immune reactions in the host body) and to avoid difficulties associated with the large size of the *COL7A1* gene [33,34]. It is obvious that the *COL7A1* cDNA size corresponds to the maximum load capacity of most viral vectors used for gene therapy (8–9 kb), which limits the effectiveness of ex vivo transduction [175].

It should be noted that one of the difficulties of editing the cell genome in vivo is the delivery of the gene editor to ESCs located on the basement membrane, which requires crossing the multilayer epidermis [22]. Many researchers are currently working to solve this problem. In particular, Keles E. et al. (2016) have proposed to use the nanoneedle-based tools to deliver the gene editor to skin cells [176].

In 2020, the European Medicines Agency granted the orphan drug status to the drug developed on the basis of CRISPR/Cas9-mediated excision of the 80th exon of the *COL7A1* gene bearing a mutation as the potential new therapy for RDEB patients [33,34].

The CRISPR/Cas9 system has demonstrated a number of advantages over the TALEN system, in particular, the high editing efficiency with significantly lower costs compared to nucleases focused on the protein structure, as well as easy production and the ability to edit any DNA site [34]. Shinkuma S. et al. (2016) edited the heterozygous mutation causing DDEB in the *COL7A1* gene using iPSCs [150]. These iPSCs were obtained from human DFs bearing the mutation by the NHEJ mechanism using TALEN and CRISPR/Cas9 in parallel. It was revealed that the CRISPR/Cas9 system performs editing approximately 2–3 times more efficiently than TALEN [177]. The disadvantage of the CRISPR/Cas9 system is non-targeted activity, when guide RNA is annealed nonspecifically and promotes the cutting of the different DNA sequences that are frequently localized in another gene and on another chromosome [35,170,178]. This activity can cause lethal changes in the edited cell; therefore, the main task after editing is to select cells which successfully passed the specific editing and bear no nonspecific mutations in other DNA sequences [168].

Nevertheless, significant progress has already been achieved in studying the possibility of using CRISPR/Cas9 tools to deliver and edit the genes for RDEB therapy [169]. The RDEB gene therapy targeted at the skin stem cells in vivo is the extremely important innovative strategy to treat the most severe hereditary human disease. In general, today, the RDEB genome editing technologies are still at the early stages of development, i.e., preclinical studies (Table 5). However, according to the obtained research results, accurate gene correction was achieved both in the primary skin cells and in iPSCs, which confirms the prospects of using this method for gene therapy of RDEB patients [3,38,129,167,179].

Finally, it should be emphasized that the modern strategies of ex vivo gene replacement therapy (already reached the stage of clinical trials) and ex vivo gene editing (still at the stage of preclinical studies) allow one to achieve a high level of correction of the *COL7A1* mutation that is the main cause of RDEB. Great hopes are connected with the gene therapy methods and approaches targeted at skin stem cells, e.g., iPSCs, which are actively studied today and can probably provide RDEB patients with an unlimited amount of autologous cells producing functional C7.

The effectiveness of in vivo transgene delivery for systemic use still remains a controversial issue, and no breakthrough in this direction has been achieved. In addition, for all gene strategies, the unresolved issue is related to the risk of developing adverse immune reactions against new C7 in patients with zero expression of this protein (Table 5). The currently conducted clinical studies include solely patients with residual C7 expression or with the absence of antibodies against the suggested protein product. Therefore, developing the strategy directed to immune tolerance to the newly produced C7 is currently one of the tasks in the field of gene therapy. Solving this problem can extend the gene therapy methods to all RDEB patients without exception.

Many issues of gene therapy for RDEB patients remain unresolved; however, indisputably, the strategies and approaches of this field of modern medicine have promising prospects, which is confirmed by the results of preclinical and clinical studies. Confidence in the undoubted progress in this area is also supported by huge achievements in the gene therapy of other hereditary diseases (clinicaltrials.gov: NCT03081715, NCT03398967, NCT03166878, NCT02793856, NCT03044743, NCT03164135) [3]. 

**Table 5 ijms-25-10270-t005:** The comparative analysis of gene therapy methods.

Approach	Status	Advantages	Disadvantages
Gene Replacement Therapy	*TETs:* phase III finished, EB-101; *TLTs*: phase I/II, EBGraft	Close access to clinical practice	Short-term effect; the risk of developing adverse immune reactions against the new C7 (patients with zero C7); the risk of transgene overexpression; invasiveness; systemic anesthesia and hospitalization; the risk of insertional mutagens
	*B-VEC:* phase III finished	Has been entered to clinical practice; being topical; possible to reuse; available to patients	The risk of developing adverse immune reactions against the new C7 (patients with zero C7)
Gene Editing	*TALEN, CRISPR/Cas9:* the stage of preclinical studies	The irreversibly changed gene nucleotide sequence in the DNA-mutated region is not bearing the risk of insertional mutagenesis; the long-term/lifelong restoration of the defective genes	The risk of developing adverse immune reactions against the new C7 (patients with zero C7); the difficulties of the delivery of the gene editor to ESCs; *TALEN* is laborious with complex cloning; *CRISPR/Cas9* has non-targeted activity

## 4. Conclusions

The significant advances in gene therapy and optimization of cell culture protocols allowed one to develop the efficient ex vivo and in vivo approaches to treat RDEB. The developed biomedical products are at different stages of processing; some of them are already used in the clinic, while the others are still at the step of laboratory and preclinical studies (Table 6).

The preclinical and clinical studies using the genetically modified Ks and DFs have demonstrated the promising potential to treat RDEB. The studies have revealed the problem related to the low number of ESCs in the skin Ks’ population obtained from RDEB patients. The studies have shown that the problem can be solved using iPSCs due to their unlimited proliferative and differentiation potential, as well as the ability to self-renew, which is extremely important to grow the cell mass from one edited cell.

The complexity of mechanisms underlying RDEB dictates the use of a combined approach to the treatment of RDEB patients to achieve optimal clinical outcome. This conclusion was confirmed by the possibilities of gene therapy in combination with cellular therapy. Thus, the *COL7A1* therapy with replacement or correction of recombinant C7 protein for healing skin wounds in combination, e.g., with MSCs or SCsP (to obtain systemic immunomodulatory and anti-inflammatory effect), can significantly reduce the severity of RDEB, and can also increase skin resistance to the damage and improve the life quality of such patients. The use of autologous patient cells in combination with the gene therapy based on using the programmable synthetic nucleases providing the more accurate approach to correct the RDEB mutations can allow one to obtain long-term (possibly lifelong) clinical effects. It is advisable to use these methods starting from the childhood of patients, i.e., before the onset of functional damages or other complications. In this case, the chances to avoid serious RDEB complications including SCC are high, preventing the development of chronic inflammation and fibrosis and significantly improving the life quality and expectancy of patients. It is known that RDEB is one of the most severe hereditary diseases with significant mortality at a young age.

In addition, the combined approach using iPSCs and the gene editing strategy provides new opportunities for the development of personalized replacement therapy methods.

Thus, the advantages achieved in cell and gene therapy result in substantial and very encouraging prospects in the treatment of RDEB patients due to the genetic determinism of this severe debilitating disease. However, the significant number of preclinical and clinical studies of the safety of this therapy is required before these innovative technologies can be introduced into clinical practice.

## Figures and Tables

**Table 3 ijms-25-10270-t003:** Clinical trials of alloMSCs for RDEB treatment.

Research Group	Number of RDEB Patients	Clinical Trial (N)	Reference
BM-MSCs			
El-Daraut M. et al. (2016)	14		[30]
Petrof G. et al. (2015)	10		[31]
Rashidghamat E. et al. (2020)	9	phase I/II, NCT02323789	[32]
Conget P. et al. (2010)	2	OMIM 226600	[29]
hUCB-MSCs			
Lee S. et al. (2021)	6	phase I/IIa, NCT04520022	[106]
ABCB5 + MSCs			
Kiritsi D. et al. (2021)	16	phase I/IIa, NCT03529877	[126]

**Table 6 ijms-25-10270-t006:** Summarized data on clinical trials of cell and gene therapy products developed for RDEB treatment.

Cell Type	Clinical Trial	Clinical Product	Administration	References
Ks	Phase I/IIa	Genetically corrected autologous epidermal grafts (ex vivo)	Epidermal graft	[54,136]
	Phase III (NCT04227106)	EB-101(ex vivo)	Epidermal graft	[53,55]
DFs	Phase II (ISRCTN67757229)	AlloDFs	Intradermal injection	[23]
	Phase II	AlloDFs	Intradermal injection	[24]
	Phase I (NCT02493816)	AutoDFs genetically corrected (ex vivo)	Intradermal injection	[25]
	Phase I/II (NCT02810951)	FCX-007: AutoDFs genetically corrected (ex vivo)	Intradermal injection	[65]
Ks and DFs	Phase I/II (NCT01874769)	EBGraft: ex vivo genetically corrected two-layer skin equivalents	Skin equivalent	[145]
BM cells	Phase I/II (NCT00478244)	Allogenic BM cells	BMT	[27]
MSCs	Phase I	Allogenic BM-MSCs	Intravenous infusion	[30]
	Phase I/II (NCT02323789)	Allogenic BM-MSCs	Intravenous infusion	[31]
	Phase I	Allogenic BM-MSCs	Intravenous infusion	[32]
	Phase I	Allogenic BM-MSCs	Intradermal injection	[29]
	Phase I/II (NCT04520022)	Allogenic hUCB-MSCs	Intravenous infusion	[104]
	Phase I/IIa (NCT03529877)	Allogenic ABCB5+ dermal MSCs	Intravenous infusion	[126]
B-VEC	Phase III (finished) (NCT04917874)	Gene product in vivo	Topical	[162,163]

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
