# Peer review of "Current Status of Biomedical Products for Gene and Cell Therapy of Recessive Dystrophic Epidermolysis Bullosa"

_ijms, 2024, doi:10.3390/ijms251910270_

Round 1

Reviewer 1 Report

Comments and Suggestions for Authors

Review of “The Current Status of Biomedical Products for Gene and Cell Therapy of Recessive Dystrophic Epidermolysis Bullosa” by Zorina et al.

Dystrophic epidermolysis bullosa (DEB) represents a significant subtype within the spectrum of epidermolysis bullosa disorders, characterized by fragile skin that blisters with minimal trauma. This review article nicely discusses emerging therapeutic strategies for Recessive Dystrophic Epidermolysis Bullosa (RDEB). The authors’ comprehensive analysis of extensive public data is commendable. To enhance the manuscript’s contribution to the International Journal of Molecular Sciences (IJMS), the following revisions are suggested:

Major comments:

  • While the language of this review has no problems with grammar, wordiness is a major issue and is an obstacle to understanding this review. It is hard to read. For example, the introduction itself is about 1000 words and can be reduced to deliver its message more concisely. Addressing this issue will advance this manuscript to a new level.
  • Other sections could be reduced by at least 15% easily by removing redundant phrases and sentences. Please improve.
  • The Introduction section does not provide details about what is the purpose of this review, what are the main aspects covered, and how this review is important in the context of the entire EB field. How is this review different from other similar reviews on the topic? Please provide a concise and clear summary.
  • Table 1 could be reinforced by providing major mutation sites in genes encoding these proteins, leading to EB.
  • Table 6 should definitely provide an outcome for each product in terms of its effectiveness, tolerability, etc.
  • The authors should use numbered list (with titles and subtitles) to clarify the structure of this massive review. For example, it feels that “The main directions in the development of RDEB cell therapy” is 1. and “Keratinocytes and dermal fibroblasts as the cellular components of RDEB therapy” is 1.1., but currently, both titles have the same level. Very confusing.
  • Given that the ultimate RDEB treatment involves gene editing of the COL7A1 mutation, the manuscript touches upon the controversial topic of embryonic and fetal genome editing. A brief discussion on the ethical implications and legal challenges is warranted.
  • This recent and highly relevant paper must be discussed: Hwang A et al. Therapies for cutaneous squamous cell carcinoma in recessive dystrophic epidermolysis bullosa: a systematic review of 157 cases. Orphanet J Rare Dis. 2024;19(1):206.

Minor comments:

  • “To date, we can talk about two” please rewrite (colloquial)
  • keratinocytes (Ks) have been abbreviated several times in different parts of the review.
  • “ "graft versus host" reaction “ – please replace with “graft-versus-host disease”

Author Response

Dear reviewer,

We sincerely thank you for providing us with the opportunity to improve our manuscript. We tried to considered most of your suggestions and comments and have made the necessary revisions to improve our manuscript. We hope that you will be satisfied by our improvements.

Comments and Suggestions for Authors

Review of “The Current Status of Biomedical Products for Gene and Cell Therapy of Recessive Dystrophic Epidermolysis Bullosa” by Zorina et al.

Dystrophic epidermolysis bullosa (DEB) represents a significant subtype within the spectrum of epidermolysis bullosa disorders, characterized by fragile skin that blisters with minimal trauma. This review article nicely discusses emerging therapeutic strategies for Recessive Dystrophic Epidermolysis Bullosa (RDEB). The authors’ comprehensive analysis of extensive public data is commendable. To enhance the manuscript’s contribution to the International Journal of Molecular Sciences (IJMS), the following revisions are suggested:

Major comments:

  • While the language of this review has no problems with grammar, wordiness is a major issue and is an obstacle to understanding this review. It is hard to read. For example, the introduction itself is about 1000 words and can be reduced to deliver its message more concisely. Addressing this issue will advance this manuscript to a new level.
  • We tried to remove unnecessary phrases.
  • Other sections could be reduced by at least 15% easily by removing redundant phrases and sentences. Please improve.
  • We also tried to reduce it.
  • The Introduction section does not provide details about what is the purpose of this review, what are the main aspects covered, and how this review is important in the context of the entire EB field. How is this review different from other similar reviews on the topic? Please provide a concise and clear summary.
  • We added briefly.
  • Table 1 could be reinforced by providing major mutation sites in genes encoding these proteins, leading to EB.
  • We decided to remove this table, focusing only on the gene/characteristics of DBE mutations (RDBE, DDBE), since the review is devoted to the treatment of this particular variant of EB. Compiling a table describing all the genetic alterations underlying epidermolysis bullosa due to the presence of several genes and the frequent absence of hot mutations or other alterations is worthy of compiling a separate review of the molecular genetic causes of this disease. We are ready to prepare a detailed review if the journal will be interesting in it. But we would like to note once again that in this review we are considering existing or developing methods of treatment, and not the causes of the disease.
  • Table 6 should definitely provide an outcome for each product in terms of its effectiveness, tolerability, etc.
  • From our point of view, this information will make the table cumbersome; in the text, all this data is presented in detail and in a structured manner.
  • The authors should use numbered list (with titles and subtitles) to clarify the structure of this massive review. For example, it feels that “The main directions in the development of RDEB cell therapy” is 1. and “Keratinocytes and dermal fibroblasts as the cellular components of RDEB therapy” is 1.1., but currently, both titles have the same level. Very confusing.
  • Corrected
  • Given that the ultimate RDEB treatment involves gene editing of the COL7A1 mutation, the manuscript touches upon the controversial topic of embryonic and fetal genome editing. A brief discussion on the ethical implications and legal challenges is warranted.
  • This review does not cover the topic of editing the genome of the embryo and fetus; the methods mentioned are considered for use starting from childhood – editing of skin stem cells in vivo and – iPSCs ex vivo. Therefore, this ethical issue does not affect our review.
  • This recent and highly relevant paper must be discussed: Hwang A et al. Therapies for cutaneous squamous cell carcinoma in recessive dystrophic epidermolysis bullosa: a systematic review of 157 cases. Orphanet J Rare Dis. 2024;19(1):206.
  • Since the review is devoted to the description of currently relevant biomedical cell and gene products, we touch upon the problem of cutaneous squamous cell carcinoma in the review indirectly - as a serious complication of RDBE, therefore we consider it inappropriate to focus on this topic, however, we were pleased to study the article and included it in the list of references as confirmation of the seriousness of this problem.

Minor comments:

  • “To date, we can talk about two” please rewrite (colloquial)
  • Corrected
  • keratinocytes (Ks) have been abbreviated several times in different parts of the review.
  • Corrected
  •  “ "graft versus host" reaction “ – please replace with “graft-versus-host disease”
  • Corrected

Reviewer 2 Report

Comments and Suggestions for Authors

The manuscript ijms-3158099 reviews the gene therapy to treat recessive dystrophic epidermolysis bullosa. The manuscript is well-written and I recommend the publication after minor revisions:

1) Does the development of carriers for gene therapy improve the treatment of recessive dystrophic epidermolysis bullosa? Please, explore it as a section in the manuscript.

2) In the Conclusion section incorporate a sentence about the Future Perspective.

3) Summarizes as a Table the main reasons that the raised gene therapy is more or less important than using CRISPR.

4) Recently there have been excellent articles reporting the improvement of gene therapy combined with the photoacoustic technique. Please, explore it.

Author Response

Dear reviewer,

We sincerely thank you for providing us with the opportunity to improve our manuscript. We tried to considered most of your suggestions and comments and have made the necessary revisions to improve our manuscript. We hope that you will be satisfied by our improvements.

Comments and Suggestions for Authors

The manuscript ijms-3158099 reviews the gene therapy to treat recessive dystrophic epidermolysis bullosa. The manuscript is well-written and I recommend the publication after minor revisions:

1) Does the development of carriers for gene therapy improve the treatment of recessive dystrophic epidermolysis bullosa? Please, explore it as a section in the manuscript.   

Since this review is devoted to the development of methods/products of gene and cell therapy, we touched on the topic indirectly, indicating that research is currently being conducted in this area; in our opinion, this is a large and serious topic that requires a separate publication.

2) In the Conclusion section incorporate a sentence about the Future Perspective.

Included.

3) Summarizes as a Table the main reasons that the raised gene therapy is more or less important than using CRISPR.

Added.

 4) Recently there have been excellent articles reporting the improvement of gene therapy combined with the photoacoustic technique. Please, explore it.

We looked at the available articles on this topic, but did not find any works on the use of these methods for the treatment of RDBE patients, all the articles we found were related to the use of these methods in the treatment of cancer patients. We are well aware that RDBE often leads to the development of oncological diseases, but we do not touch upon this aspect in this work, as we believe that it requires separate consideration.

Round 2

Reviewer 1 Report

Comments and Suggestions for Authors

all comments were addressed